# The Gut Microbiota of Young Asian Elephants with Different Milk-Containing Diets

**DOI:** 10.3390/ani13050916

**Published:** 2023-03-02

**Authors:** Chengbo Zhang, Junmin Chen, Qian Wu, Bo Xu, Zunxi Huang

**Affiliations:** 1Engineering Research Center of Sustainable Development and Utilization of Biomass Energy, School of Life Sciences, Yunnan Normal University, Kunming 650500, China; 2Key Laboratory of Yunnan Provincial Education Department for Plateau Characteristic Food Enzymes, Yunnan Normal University, Kunming 650500, China

**Keywords:** Asian elephant, intestinal microbiota, diet, elephant milk, goat milk

## Abstract

**Simple Summary:**

Insufficient maternal milk is one of the important reasons for the low survival rate of young Asian elephants. Finding the optimal break milk supplementation for young Asian elephants is a matter of urgency. In our study, we investigated the microbiomes of young Asian elephants on different milk-containing diets (elephant milk only, elephant milk–plant mixed feed, and goat milk–plant mixed feed). Our results suggested that goat milk is not suitable for young elephants, and yak milk may be an ideal source of supplemental milk for Asian elephants.

**Abstract:**

Evaluating the association between milk-containing diets and the microbiomes of young Asian elephants could assist establishing optimal breast milk supplementation to improve offspring survival rates. The microbiomes of young Asian elephants on different milk-containing diets (elephant milk only, elephant milk–plant mixed feed, and goat milk–plant mixed feed) were investigated using high-throughput sequencing of 16S rRNA genes and phylogenetic analysis. Microbial diversity was lower in the elephant milk-only diet group, with a high abundance of *Proteobacteria* compared to the mixed-feed diet groups. *Firmicutes* and *Bacteroidetes* were dominant in all groups. *Spirochaetae*, *Lachnospiraceae*, and *Rikenellaceae* were abundant in the elephant milk–plant mixed-feed diet group, and *Prevotellaceae* was abundant in the goat milk–plant mixed-feed diet group. Membrane transport and cell motility metabolic pathways were significantly enriched in the elephant milk–plant mixed-feed diet group, whereas amino acid metabolism and signal transduction pathways were significantly enriched in the goat milk–plant mixed-feed diet group. The intestinal microbial community composition and associated functions varied significantly between diets. The results suggest that goat milk is not suitable for young elephants. Furthermore, we provide new research methods and directions regarding milk source evaluation to improve elephant survival, wellbeing, and conservation.

## 1. Introduction

The Asian elephant (*Elephas maximus*) is a large phytophagous mammal that is mainly found in the Xishuangbanna region of Yunnan Province, China, south of 24.6° north latitude, and in parts of south and southeast Asia [1]. The Asian elephant is a Class I protected wildlife species in China and is listed as endangered by the International Union for Conservation of Nature Red List of Threatened Species™ [2,3]. Furthermore, these elephants are in Appendix I of the Convention on International Trade in Endangered Species of Wild Fauna and Flora [4]. There are only approximately 300 Asian elephants left in China [5].

Although the Asian elephant population has rebounded after years of effort, its survival rate still requires improvement. Approximately 25.6% of elephant calves in Myanmar reportedly die before they reach 5 years of age, with a quarter of these deaths attributed to insufficient maternal milk or the inability of the calves to receive the milk properly [6]. Similarly, in wild African elephant populations, an average of 19% of young elephants die before 5 years of age, with a proportion of these deaths attributed to maternal difficulties regarding meeting nursing needs [7]. During droughts, maternal elephants struggle to maintain milk production, when the metabolic demands of young male elephants are greater, making it difficult for maternal elephants to meet their needs. Thus, young male elephants are more likely to die [8]. A major reason for the high mortality rate of elephant calves in zoos, especially in Asia, is the refusal of mothers to nurse their young, resulting in the need for manual intervention to feed the young [9,10]. Inadequate maternal milk in Asian elephants results in the poor survival rate of young elephants, and currently, staff at the Xishuangbanna Asian Elephant Sanctuary are using goat milk to supplement the feeding of rescued infants and young elephants.

The large number of microbial communities present in the gastrointestinal tract of animals constitute the microbiota, which contribute to host nutrient acquisition and immune regulation [11,12] and assist in maintaining host homeostasis in response to environmental changes [13,14,15]. Diet, especially early nutrition, influences the composition and metabolic activity of the gut microbial community and is a key factor in the growth and healthy development of newborn elephants [16,17]. Breastfeeding is considered an influential driver of the gut microbiota composition during infancy, potentially affecting the function thereof [18]. The gut microbiota early in life is associated with physiological development, and early gut microbiota is involved in a range of host biological processes, particularly immunity, cognitive neurodevelopment, metabolism, and infant health [19,20].

Early foods can promote the survival rate of infant and young elephants; therefore, it is vital to study the effects of different foods, especially different kinds of milk on the gut microbiota of infant and young elephants. In this study, the gut microbiota composition and function of young elephants fed an elephant milk-only diet, elephant milk–plant mixed-feed diet, and goat milk–plant mixed-feed diet were analyzed using 16S rRNA gene high-throughput sequencing technology. Although there have been studies regarding the use of non-breast milk dairy products for feeding endangered wildlife (e.g., Siberian tigers [16]), only few studies on the gut microbiota of Asian elephants on diets containing goat milk exist. To the best of our knowledge, this study is the first to describe the composition and function of the gut microbiota of young elephants fed a goat milk diet.

## 2. Materials and Methods

### 2.1. Fecal Sample Collection

In March 2019, we collected fresh feces from eight young Asian elephants with different milk-containing diets at Wild Elephant Valley in Xishuangbanna: three in the elephant milk diet-only group (BF1, BF2, and BF2; they are healthy, aged about 6 months, and can freely shuttle below the abdomen of adult female elephants); three in the elephant milk-plant mixed feeding group (BPM1, BPM2, and BPM3; they are healthy, more than one year old, and tall to the base of the forelegs of adult female elephants); and two in the goat milk–plant mixed feeding group (GPM1 and GPM2; they are healthy, more than three year old, and height slightly higher than the previous group). The detailed sampling method was as follows [21]: young elephants were accompanied by the breeder until defecation, samples were collected immediately from the center of fresh feces with sterile tweezers, placed in sterile centrifuge tubes, and stored in liquid nitrogen. Samples were transported in liquid nitrogen, and then stored at −80 °C until DNA extraction.

### 2.2. Genomic DNA Extraction, Gene Amplification and High-Throughput Sequencing

Microbial genetic DNA was extracted from eight fecal samples using the EZNA^®^ Soil DNA Kit (Omega, GA, USA) following the steps in the kit instructions. DNA quality and quantity were assessed using a 1% agarose gel and a NanoDrop 2000 spectrophotometer (Thermo Scientific, Wilmington, DE, USA). The hypervariable region V3-V4 of the bacterial 16S rRNA gene was amplified with the primer pair 338F (5′-ACTCCTACGGGAGGCAGCAG-3′) and 806R (5′-GGACTACHVGGGTWTCTAAT-3′) using an ABI GeneAmp 9700 PCR thermal cycler (Appliedbiosystems, Foster City, CA, USA). The PCR mix consisted of 4 μL of 5× TransStart FastPfu buffer, 2 μL of 2.5 mM dNTP, 0.8 μL each of 5 μM forward and reverse primers, 0.4 μL of TransStart FastPfu DNA polymerase, 10 ng of template DNA and ddH2O up to 20 μL. PCR amplification was performed in triplicate under the following conditions: 95 °C for 3 min, followed by 30 cycles of 95 °C for 30 s, 55 °C for 30 s, and 72 °C for 45 s, and a final extension at 72 °C for 10 min. Purified amplicons were pooled in equimolar aliquots and then sequenced on the Illumina MiSeq platform (Illumina, San Diego, CA, USA) to obtain paired-end reads [22].

### 2.3. Sequencing Data Processing

Raw 16S rRNA gene sequencing reads were demultiplexed and quality-filtered using fastp version 0.20.0 [23] and then merged using FLASH version 1.2.7 [24]. Stringent criteria were established for quality. Three hundred-base pair reads were truncated at any site that received an average quality score <20 over a 50 bp sliding window. Truncated reads shorter than 50 bp and reads with ambiguous characters were discarded. Sequences required an overlap larger than 10 bp for assembly, and the maximum mismatch ratio of the overlap region was 0.2. Reads that could not be assembled were discarded. Samples were distinguished by barcodes and primers, and the sequence direction was adjusted accordingly. Exact barcode matching was required, and a mismatch of two nucleotides in primer matching was allowed.

Operational taxonomic units (OTUs) with a 97% similarity cutoff [25,26] were clustered using UPARSE version 7.1 [25]; chimeric sequences were identified and removed. Taxon assignments for each representative OTU sequence were determined using RDP Classifier version 2.2 [27] with the 16S rRNA gene database (Silva v138) with a confidence threshold of 0.7.

### 2.4. Data Analysis and Statistical Methods

To investigate the similarity and difference relationship of microbial community structure among different milk-containing diet groups, sample-level clustering analysis was performed using UPGMA method based on the average Bray_curtis distance matrix among groups. Alpha diversity indices including Chao1 index, Shannon index, and Pielou index were calculated using software mothur (version 1.30.2, http://www.mothur.org/wiki/Schloss_SOP#Alpha_diversity, accessed on 23 April 2019), and difference tests between multiple groups were performed using Welch’s *t*-test. The Kruskal–Wallis H test was applied to detect species that exhibited differences in abundance in the microbial communities between groups. In addition, functional prediction results were obtained using PICRUSt2, and the difference significance was detected using the Kruskal–Wallis H test.

## 3. Results

### 3.1. Unweighted Pair Group Method with Arithmetic Mean Hierarchical Clustering Analysis

At the family and genus levels, the samples were analyzed using hierarchical clustering based on the unweighted pair group method with arithmetic mean (UPGMA) cluster analysis method (Figure 1), which indicated that the samples were clearly clustered into two groups: the elephant milk diet group (BF1, BF2, and BF2) and the milk–plant mixed-feed diet group (remaining samples). The milk–plant mixed-feed diet group was clearly further divided into two groups according to the type of supplemented milk: the elephant milk–plant mixed-feed diet group (BPM1, BPM2, and BPM3) and the goat milk–plant mixed-feed diet group (GPM1 and GPM2). These results exhibited that the gut microbial community composition of young elephants in the elephant milk-only diet group and that of young elephants in the milk–plant mixed-feed diet groups differed clearly. Moreover, the gut microbial community composition of young elephants in the elephant milk-only diet group and that of young elephants in the goat milk–plant mixed-feed diet group also differed significantly.

### 3.2. Alpha Diversity Analysis

An α-diversity test was performed to evaluate the differences in the gut microbial community between the three groups at the family level (Figure 2). Consequently, the richness index (Chao1) and diversity index (Shannon) were significantly different between the three groups (*p* < 0.05, Figure 2A,B). The richness and diversity indices of the milk–plant mixed-feed diet groups were significantly higher than those of the elephant milk-only diet group (*p* < 0.05), which was consistent with the richness of dietary diversity in the milk–plant mixed-feed diet groups. In addition, the Shannon and Pielou indices were significantly higher in the elephant milk–plant mixed-feed diet group than in the goat milk–plant mixed-feed diet group (*p* < 0.05, Figure 2B,C). These findings suggested that supplementation with elephant milk in young elephants resulted in a more diverse and homogeneous gut bacterial community than supplementation with goat milk, and supplementation with goat milk may lead to a highly dominant bacterial taxon in the gut environment of young elephants.

### 3.3. Community Composition

Firmicutes and Bacteroidetes represented the dominant phyla in young elephant guts, which was consistent with the dominant phyla in the gut microbiota of adult Asian elephants (Figure 3) [28]. The young elephant intestinal microbiota in the elephant milk-only diet group (BF1, BF2, and BF3) contained a high abundance of *Proteobacteria*, averaging around approximately 17.3% (Figure 3). The elephant milk–plant mixed-feed diet group (BPM1, BPM2, and BPM3) had a higher abundance of *Spirochaetae* (approximately 8.8%), *Fibrobacteria* (approximately 3.8%), and *Verrucomicrobia* (approximately 3.6%) compared to the elephant milk-only diet group (Figure 3). The BPM1 group had a relatively higher intake of elephant milk and, correspondingly, higher abundance of Proteobacteria, while BPM2 and BPM3, which had lower intakes of elephant milk, had an extremely low abundance of Proteobacteria, indicating that elephant milk is closely related to Proteobacteria levels. The goat milk–plant mixed-feed diet group (GPM1 and GPM2) contained nearly no *Proteobacteria*, *Spirochaetae*, and *Fibrobacteria* (Figure 3), and the considerably low abundance of Proteobacteria indicated that elephant milk is closely related to the abundance of this bacterium. In addition, Synergistetes were abundant in the intestinal microbiota of young elephants in the goat milk–plant mixed-feed diet group compared to the other groups (Figure 3).

At the family level, the intestinal bacteria of young elephants in the elephant milk-only diet group consisted mainly of *Bacteroidaceae*, *Enterobacteriaceae*, *Ruminococcaceae*, and *Lachnospiraceae*, accounting for >75% of intestinal bacteria (Figure 1A). The intestinal bacteria of young elephants in the elephant milk–plant mixed-feed diet group consisted mainly of *Lachnospiraceae*, *Ruminococcaceae*, *Rikenellaceae*, *Spirochaetaceae*, and *Prevotellaceae*, accounting for >70% of intestinal bacteria (Figure 1A). BPM1, who consumed a large amount of elephant milk, had an abundance of *Enterobacteriaceae*, suggesting that *Enterobacteriaceae* levels are closely related to the elephant milk consumed by young elephants. The intestinal bacteria of young elephants in the goat milk–plant mixed-feed diet group consisted mainly of *Ruminococcaceae*, *Lachnospiraceae*, *Prevotellaceae*, and *Synergistaceae*, accounting for approximately 60% of intestinal bacteria (Figure 1A).

### 3.4. Differential Microbiota Analysis

At the family level, differential microbiota analysis of young elephants (Figure 4) revealed that *Rikenellaceae*, *Spirochaetaceae*, *Fibrobacteraceae*, and *Bacteroidales*_UCG-001 were significantly enriched in the elephant milk–plant mixed-feed diet group (*p* < 0.05). These bacterial taxa belong to the lignocellulose-degrading bacterial phyla commonly encountered in the gastrointestinal tracts of animals, such as *Bacteroidetes*, *Spirochaetes*, and *Fibrobacteres*, suggesting that elephant milk enriches lignocellulose-digesting bacterial groups in the intestinal tract of young elephants, facilitating the transition from an elephant milk diet to a plant-based diet. *Prevotellaceae*, *Synergistaceae*, and *Christensenellaceae* were significantly enriched in the goat milk–plant mixed-feed diet group (*p* < 0.05). This indicated that there was a significant difference in the effect of elephant and goat milk supplementation in the diet on the intestinal microbiota of young elephants.

### 3.5. Function Predictive Analysis

Predictive analysis of the intestinal microbiota function in young elephants revealed differences in microbial community functions between different milk-containing diet groups (Figure 5). Carbohydrate and cofactor metabolism, vitamins, and glycan biosynthesis and metabolism were significantly more enriched in the elephant milk-only diet group compared to the mixed-feed diet group (*p* = 0.044). These function enrichments were beneficial to infant elephant growth and development. The enrichment of nucleotide metabolism (*p* = 0.044) and biosynthesis of other secondary metabolites (*p* = 0.044) were significantly higher in the goat milk–plant mixed-feed diet group compared to that of the elephant milk-only diet group, indicating that secondary metabolic pathways occurred during food digestion in the goat milk–plant mixed-feed diet group. The other amino acid metabolic (*p* = 0.030), transformation (*p* = 0.046), transcriptional (*p* = 0.020), replication and repair (*p* = 0.030), endocrine system (*p* = 0.044), and cell growth and death metabolic (*p* = 0.030) pathways were also significantly more enriched in the elephant milk–plant mixed-feed diet group than in the elephant milk-only diet group. The significant enrichment of these functions reflected strong metabolism and good growth and development of the young elephants in this group, indicating that the elephant milk–plant mixed-feed diet promoted the transition of young elephants from an elephant milk-based diet to a plant-based diet. In the elephant milk–plant mixed-feed diet group, enrichment of the membrane transport pathway (*p* = 0.044) and cell motility pathway (*p* = 0.044) was significantly higher in the elephant milk–plant mixed-feed diet group than in the goat milk–plant mixed-feed diet group. Meanwhile, the energy metabolic (*p* = 0.044), amino acid metabolic (*p* = 0.044), and signal transduction (*p* = 0.025) pathways were significantly more enriched in the goat milk–plant mixed-feed diet group than in the elephant milk–plant mixed-feed diet group. These results suggested that supplementation of the host’s diet with milk from different sources led to changes in the functional structure of the gut microbiota in Asian elephants.

### 3.6. Composition Comparison of Different Kinds of Milk

There were significant differences in the composition and function of the gut microbiota between the elephant milk diet groups and the goat milk diet group of young elephants (Figure 4 and Figure 5). Moreover, there was a close correlation between the host’s diet and their gut microbiota [29,30], where diet may have represented the main reason for these differences. Previous studies have shown significant differences in the nutrient composition of Asian elephant milk [6,10,31,32] compared to goat milk [33,34]. In the Asian elephant milk, the total solids (17.56–19.60%), protein (3.30–5.23%), and milk fat (7.70–8.30%) content were significantly higher than those in the goat milk (11.53–13.00%, 3.17–3.75%, and 3.95–4.25% for total solids, protein, and fat contents, respectively), while the water content (81.90–82.44%) was significantly lower than that of the goat milk (88.00%) (Table 1). The differences in the gut microbiota composition and function between the mixed-feed diet groups in this study may be mainly due to the differences in the nutrient composition and content between elephant milk and goat milk.

Comparisons of the nutrient composition and content of different kinds of milk and Asian elephant milk have been conducted in previous studies [35,36,37,38]. The nutritional composition and content of yak milk [35,36] was similar to that of Asian elephant milk (Table 1). Water, total solids, protein, milk fat, ash, and lactose accounted for 83.74%, 16.60–18.52%, 4.68–5.41%, 6.72–8.18%, 0.72–1.19%, and 4.40–5.10% of yak milk, respectively (Table 1). Although there has been no study on the intestinal microbiota of Asian elephants supplemented with yak milk, the similarity between the composition and content of yak milk and Asian elephant milk suggests that yak milk may represent a viable choice of milk compared to goat milk for the supplementation of rescued young Asian elephants.

## 4. Discussion

Asian elephants are endangered wild animals, and there are few milk-drinking young elephants. Although the number of samples in each group in this study is insufficient, this is all the samples that could be collected in Xishuangbanna region at that time. Here, the diversity of gut microbial communities of young elephants differed significantly between different milk-based diet groups, reflecting the various effects that these diets may have on the growth and development of young elephants. The richness (Chao 1 index) and diversity (Shannon index) of human intestinal microbiota are crucial indicators of health [39]. Claesson et al. [40] reported that preterm infants with necrotizing colitis have a significantly lower diversity of fecal microbiota compared to those without the disease, and young children with lower gut microbiota diversity are at higher risk of developing allergic diseases later in life. Thus, the greater the gut microbiota richness and diversity, the more likely it is that the nutritional status and health of the host will be good. In this study, the elephant milk–plant mixed-feed diet group had higher intestinal microbiota diversity compared to the goat milk–plant mixed-feed diet group; therefore, although it is feasible to feed goat milk to young elephants, these results suggest that more suitable milk sources should be identified to serve as appropriate elephant milk supplementation for Asian elephants.

Firmicutes and Bacteroidetes were the dominant phyla in all three groups, which is consistent with the results of Ilmberger et al. [41], and are also the dominant phyla in the adult Asian elephant gut microbiota [21]. Intestinal Firmicutes have many genes encoding fermentable dietary fiber proteins, which can also interact with the intestinal mucosa, contributing to the stability of the host’s internal environment [42]. Bacteroidetes are the main drivers of plant biomass degradation in Asian elephants [21,28,41]. These two bacterial taxa are indispensable for Asian elephants, as they assist plant digestion for energy acquisition. In the goat milk–plant mixed-feed diet group, the dominant phyla in the gut remained Firmicutes and Bacteroidetes, indicating that the use of goat milk to feed young Asian elephants could maintain the stability of the dominant phyla in the intestinal microbiota, allowing digestion and energy acquisition from food. The abundance of *Spirochaetae* in the intestinal microbiota of young Asian elephants was higher in the elephant milk–plant mixed-feed diet group compared with the goat milk–plant mixed-feed diet group. *Spirochaetae* are associated with the cell motility pathway, which is required by intestinal microbiota to actively contact their substrates and facilitate the biochemical reactions of the substrates [43,44]. This suggests that goat milk is not the most suitable supplement for elephant milk. In addition, *Lachnospiraceae* were more abundant in young Asian elephants in the elephant milk–plant mixed-feed diet group compared to in the goat milk–plant mixed-feed diet group, and are closely associated with host mucosal integrity, bile acid metabolism, and polysaccharide catabolism [45]. The low *Lachnospiraceae* abundance in the goat milk–plant mixed–feed diet group further suggested that goat milk may not be the best choice for feeding young Asian elephants. The abundance of *Prevotellaceae* and *Rikenellaceae* was higher in the mixed-feed diet groups than in the elephant milk-only diet group. A low abundance of *Rikenellaceae* and a high abundance of *Prevotellaceae* have been associated with obesity [46,47]. Therefore, the lower abundance of *Rikenellaceae* and higher abundance of *Prevotellaceae* in the goat milk–plant mixed-feed diet group compared to the elephant milk–plant mixed-feed diet group suggest that goat milk–plant mixed feeding may cause obesity in Asian elephants. In turn, this could lead to a potential risk of obesity-related diseases in Asian elephants. *Synergistaceae* encode multiple pathways that may be associated with the metabolism of diet-generated compounds [48], and these are predicted to be key factors in dietary detoxification in herbivores. In this study, *Synergistaceae* were significantly enriched in the goat milk–plant mixed-feed diet group, which was consistent with the significant enrichment of biosynthesis of other secondary metabolites in this group. This was likely due to the excess of secondary metabolism occurring during food digestion in this group. Meanwhile, the reason behind excess secondary metabolism, caused by the supplementation of goat milk or the presence of specific components in the foraged plants, requires further elucidation.

Recent studies on the relationship between breast milk and the gut microbiota have revealed a correlation between milk composition and gut microbiota in infants [31], and that milk composition varies by mammalian species [49,50]. The composition and content of Asian elephant [5,10,31,32] and goat milk [33,34] differ significantly. Asian elephant milk is richer in nutrients than goat milk, which may have been the main reason for the difference in the composition and function of the gut microbiota between the elephant milk–plant mixed-feed diet group and the goat milk–plant mixed-feed diet group. Nutrient composition analysis and the content of yak milk [35,36] indicates that it is similar to Asian elephant milk. Furthermore, through the study of yak milk on retinoic acid-induced osteoporosis in mice, it was found that yak milk could improve bone quality and microstructure to promote bone health [51]. The study of Zhang Wei et al. showed that yak milk could improve endurance capacity and relieve fatigue [52]. It is reported that yak dairy products seem to be particularly rich in functional and bioactive ingredients, which may play a role in maintaining the health of nomadic peoples [53]. Nutritional composition analysis of yak milk and its advantages in other animals suggested that yak milk may be an ideal source of supplemental milk for Asian elephants, compared to goat milk.

## 5. Conclusions

By studying the gut microbiome of Asian elephants on different milk-containing diets, it revealed the fact that the diet supplemented with goat milk diet seems not to be the most indicated to young elephants, and the composition and function of the gut microbiota of young elephants on a supplemented goat milk diet were also revealed for the first time, which were compared with those on an elephant milk diet only and an elephant milk–plant mixed-feed diet. This study presents a breakthrough in a new research area, the gut microbiome, regarding the serious problem of a low survival rate of infant and young elephants due to insufficient breast milk. Furthermore, we demonstrate the importance of finding a more suitable supplemental or alternative source of breast milk for Asian elephants. We believe that, in the future, with the help of wildlife gut microbiome analysis, the best supplemental or alternative sources of milk can be identified for other endangered wildlife infants and young to enhance the wellbeing of wildlife and relieve the threat to survival caused by insufficient breast milk.

## Figures and Tables

**Figure 1 animals-13-00916-f001:**
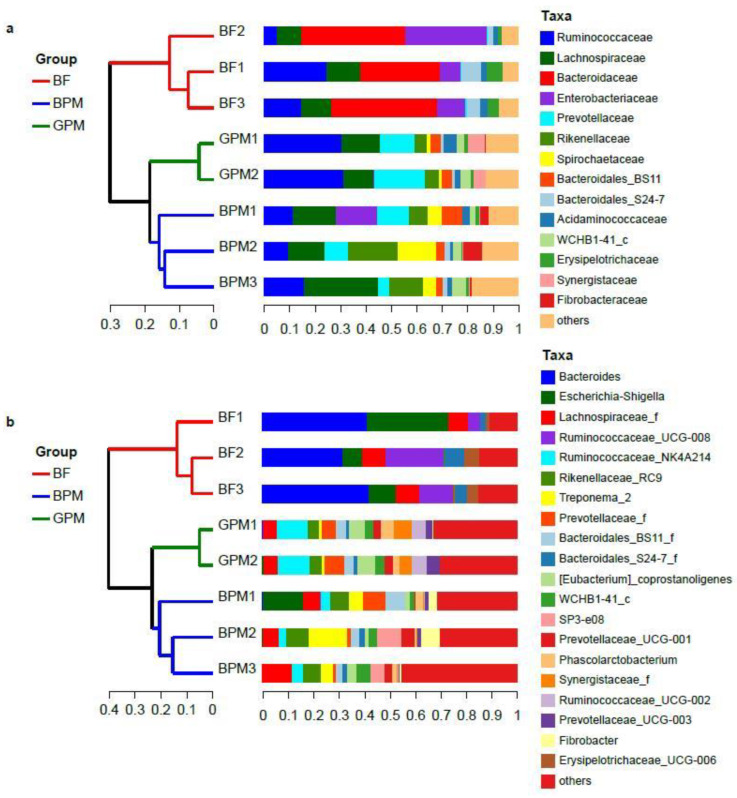
(**a**) Hierarchical clustering tree of species at the family level obtained from samples based on the unweighted pair group method with arithmetic mean (UPGMA) clustering analysis method; (**b**) species-level hierarchical clustering trees at the genus level obtained from samples based on the UPGMA cluster analysis method. BF1, BF2, and BF2 refer to the elephant milk-only diet group, BPM1, BPM2, and BPM3 refer to the elephant milk–plant mixed-feed diet group, and GPM1 and GPM2 refer to the goat milk–plant mixed-feed diet group.

**Figure 2 animals-13-00916-f002:**
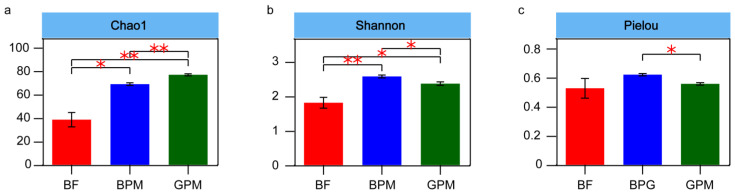
Analysis of the α-diversity of microbial communities. (**a**) Chao 1 index; (**b**) Shannon index; and (**c**) Pielou index. BF refers to the elephant milk-only diet group, BPM refers to the elephant milk–plant mixed-feed diet group, and GPM refers to the goat milk–plant mixed-feed diet group. * refers to *p* < 0.05, ** refers to *p* < 0.01.

**Figure 3 animals-13-00916-f003:**
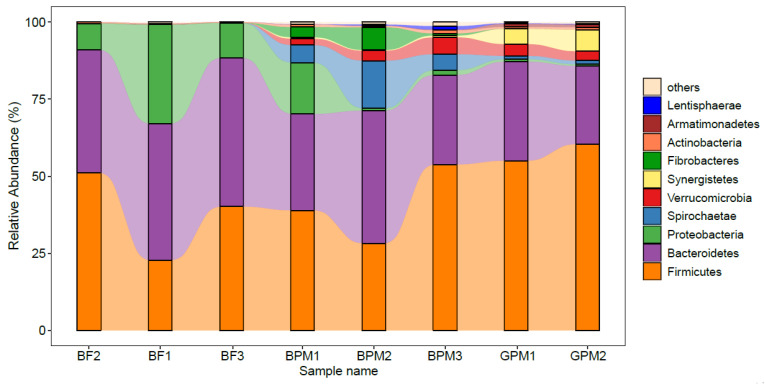
Gut microbial community composition of young elephants at the phylum level.

**Figure 4 animals-13-00916-f004:**
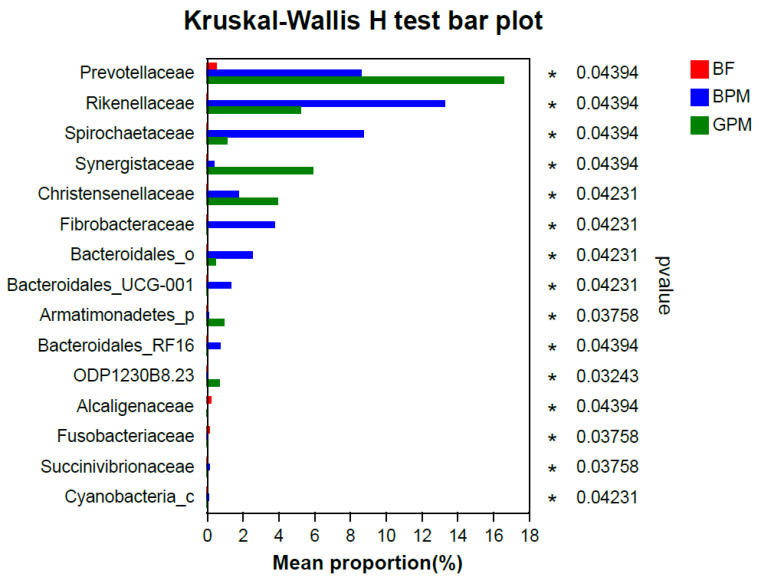
Differential gut microbiota analysis of young elephants. *: Significant difference.

**Figure 5 animals-13-00916-f005:**
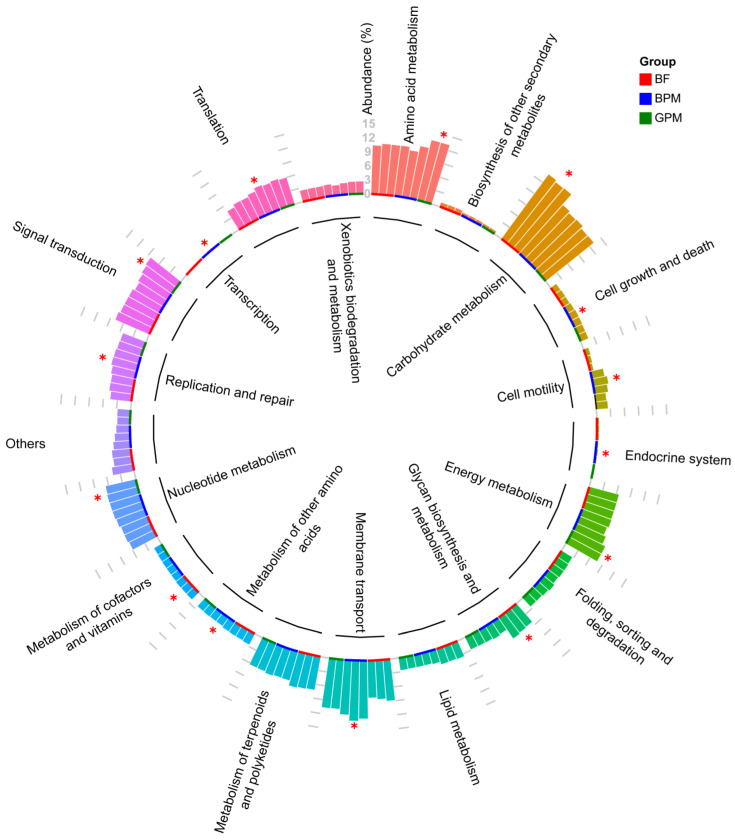
Predictive analysis of the gut microbial function in young elephants. * Metabolic pathways that were significantly different between groups (*p* < 0.05).

**Table 1 animals-13-00916-t001:** Comparison of the nutrient composition and content of various kinds of milk.

Nutritional Composition	Asian Elephant Milk	Goat Milk	Yak Milk	Camel Milk	Sheep Milk
Moisture	81.90–82.44%	88.00%	83.74%	85.0–90.0%	88%
Total solids	17.56–19.60%	11.53–13.00%	16.60–18.52%	9.95–16.31%	9.60–9.95%
Protein	3.30–5.23%	3.17–3.75%	4.68–5.41%	3.00–4.19%	3.18–3.29%
Milk fat	7.70–8.30%	3.95–4.25%	6.72–8.18%	4.43–6.26%	6.79–7.20%
Ash	0.40–0.87%	0.79%	0.72–1.19%	0.68–0.71%	0.93–0.96%
Lactose	3.00–4.90%	4.20–4.82%	4.40–5.10%	4.25–5.43%	5.49–5.70%

## Data Availability

The data supporting the results of this study can be found in the manuscript. Raw sequence data obtained in this Raw sequence data obtained in this study have been deposited in the Genome Sequence Archive of the BIG Data Center of the Chinese Academy of Sciences (GSA: CRA009476) and are publicly accessible at https://ngdc.cncb.ac.cn (accessed on 1 March 2023).

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
