# Peer review of "The Gut Microbiota of Young Asian Elephants with Different Milk-Containing Diets"

_animals, 2023, doi:10.3390/ani13050916_

Round 1

Reviewer 1 Report

The manuscript "The gut microbiota of young Asian elephants with different milk-containing diets" focus an important subject related to a rare and endangered species with low survival rate of young individuals. 

The Abstract is correct and elucidates the content of the manuscript.  

The introduction section is satisfactory. 

The materials and methods seem adequate. Line 86 - young elephants instead of your… 

The results are sound, even thought the low number of young elephants in each group. 

The discussion is correct, but the authors should emphasize the low number of young elephants in each group… 

The manuscript should be published as it is vital to study the effects of different foods on the gut microbiota of infant and young elephants.

The main question of this manuscript is if different milk-containing diets (elephant milk only, elephant milk–plant mixed feed, and goat milk–plant mixed feed) influence the gut microbioma of young Asian elephants. 

 I consider the topic original in the field, since is the first report regarding the influence of goat milk–plant mixed feed in gut microbiota constitution and functionality of young elephants, relating with elephant milk only. It brings useful information since goat milk seems not be the most adequate to that species. 

The authors should considerer increase the number of animals/samples in the study. Since they consider that Yak milk is the most similar in composition to elephants’ milk, it would be interesting the study the microbioma of young elephants feed with Yak milk. 

The conclusions should be more objectively related to the fact that supplemented goat milk diet seems not be the most indicated to young elephants.

Reviewer 2 Report

Dear Authors

This manuscript is very significant important to elephant care either for wild and captive,

(I) I understand that all fecal samples from elephants in same age, correct?

(II)Would be nice if authors can explain and comparison elephant performance, health between group of BF-BPM and GPM in M&M or Results ?

(III)Resulting of microbial founding among 3 groups of milk feeding can be use in discussion part, however authors might need to re-shape and explain about relative of each microbial to feeding type and elephant health/performance, to make more clear for readers.

(IV) Suggesting that Yak milk might become a better solution by comparison of nutrition composition might not enough, so authors might need to show advantage for Yak milk using in other aspect such in other animal spp.

Finally, this manuscript will be a good for elephant care when it publish.

Cheers,      
